# The Impact of Expert Pathology Review and Molecular Diagnostics on the Management of Sarcoma Patients: A Prospective Study of the Hellenic Group of Sarcomas and Rare Cancers

**DOI:** 10.3390/cancers16132314

**Published:** 2024-06-24

**Authors:** Stefania Kokkali, Ioannis Boukovinas, Eelco de Bree, Anna Koumarianou, Vassilis Georgoulias, Anastasios Kyriazoglou, Nikolaos Tsoukalas, Nikolaos Memos, John Papanastassiou, Anastasia Stergioula, Konstantinos Tsapakidis, Konstantia Loga, Jose Duran-Moreno, Panagiotis Papanastasopoulos, Nikolaos Vassos, Vasileios Kontogeorgakos, Ilias Athanasiadis, Luiza Mahaira, Efthymios Dimitriadis, Dionysios J. Papachristou, George Agrogiannis

**Affiliations:** 1Oncology Unit, 2nd Department of Medicine, Medical School, Hippocratio General Hospital of Athens, National and Kapodistrian University of Athens, V. Sofias 114, 11527 Athens, Greece; 2Oncology Department, Bioclinic of Thessaloniki, 54622 Thessaloniki, Greece; ibouk@bioncology.gr; 3Department of Surgical Oncology, University Hospital of Heraklion, 71110 Heraklion, Greece; debree@med.uoc.gr; 4Hematology Oncology Unit, Fourth Department of Internal Medicine, School of Medicine, National Kapodistrian University of Athens, 12462 Athens, Greece; akoumari@yahoo.com; 5Hellenic Oncology Research Group (HORG), 11526 Athens, Greece; georgulv@otenet.gr; 6Medical Oncology Unit, Department of Internal Medicine, School of Medicine, National and Kapodistrian University of Athens, 12462 Athens, Greece; tassoskyr@gmail.com; 7Department of Oncology, 401 General Military Hospital of Athens, 11525 Athens, Greece; tsoukn@yahoo.gr; 82nd Department of Surgery, School of Medicine, Aretaieion Hospital, National and Kapodistrian University of Athens, 11528 Athens, Greece; nikolaosmemos@gmail.com; 9Department of Orthopedic Oncology, “Agioi Anargyroi” General Hospital, 14564 N.Kifisia, Greece; ioannis.papanastassiou@gmail.com; 10Department of Radiation Oncology, “Iaso” Hospital, 15123 Marousi, Greece; nastasia_gr@hotmail.com; 11Department of Tomotherapy-Stereotactic Radiosurgery “Iatropolis”, 15231 Chalandri, Greece; 12Department of Medical Oncology, University Hospital of Larissa, 41334 Larissa, Greece; tsapakidisk@yahoo.com; 13Department of Medical Oncology, School of Medicine, Faculty of Health Sciences, Papageorgiou Hospital, Aristotle University of Thessaloniki, 56429 Thessaloniki, Greece; ntinaloga@yahoo.com; 14Hellenic Group of Sarcoma and Rare Cancers, G. Theologou 5, 11471 Athens, Greece; duranmoreno.jose@gmail.com; 15Oncology Unit, ST Andrews General Hospital of Patras, 26332 Patras, Greece; ppapanastasopoulos@nhs.net; 16Division of Surgical Oncology and Thoracic Surgery, Mannheim University Medical Center, University of Heidelberg, 68167 Mannheim, Germany; nikolaos.vassos@umm.de; 17Department of Surgical Oncology, Athens Medical Center, 15125 Athens, Greece; 181st Department of Orthopaedic Surgery, School of Medicine, National and Kapodistrian University of Athens, 12462 Athens, Greece; vaskonto@gmail.com; 19Oncology Department, Hygeia Athens Private Hospital, 15123 Maroussi, Greece; iathanasiadis@hygeia.gr; 20Department of Genetics, Saint Savvas Cancer Hospital, 11522 Athens, Greece; genetics@agsavvas-hosp.gr (L.M.); dthimios@0083.syzefxis.gov.gr (E.D.); 21Unit of Bone and Soft Tissue Studies, Department of Histology and Histopathology, School of Medicine, University of Patras, 26504 Patras, Greece; papachristoudj@med.upatras.gr; 22Department of Pathology, School of Medicine, National and Kapodistrian University of Athens, 11527 Athens, Greece; agrojohn@med.uoa.gr

**Keywords:** sarcoma, soft tissue sarcoma, bone sarcoma, diagnostics, pathology, molecular test, molecular methods, next-generation sequencing

## Abstract

**Simple Summary:**

Sarcomas represent a group of rare tumors consisting of more than 100 different entities. A precise diagnosis is crucial to optimal treatment. In addition, a significant proportion of sarcomas are characterized by a specific genetic abnormality. This study was designed by the Hellenic Group of Sarcoma and Rare Cancers to assess the effect of expert pathology review, coupled with the application of molecular methods to detect genetic abnormalities, on the diagnosis and management of sarcoma patients in Greece. We found that histological diagnosis review by a sarcoma pathologist led to modifications in diagnosis in almost one-third of patients, resulting in modifications in management in 14% of patients. The use of molecular tests led to modifications in diagnosis in 26% of patients, resulting in modifications in management in almost 11% of patients. Our study highlights the importance of expert pathologists, assisted in some cases by molecular methods, in sarcoma diagnosis and treatment.

**Abstract:**

Precise classification of sarcomas is crucial to optimal clinical management. In this prospective, multicenter, observational study within the Hellenic Group of Sarcoma and Rare Cancers (HGSRC), we assessed the effect of expert pathology review, coupled with the application of molecular diagnostics, on the diagnosis and management of sarcoma patients. Newly diagnosed sarcoma patients were addressed by their physicians to one of the two sarcoma pathologists of HGSRC for histopathological diagnostic assessment. RNA next-generation sequencing was performed on all samples using a platform targeting 86 sarcoma gene fusions. Additional molecular methods were performed in the opinion of the expert pathologist. Therefore, the expert pathologist provided a final diagnosis based on the histopathological findings and, when necessary, molecular tests. In total, 128 specimens from 122 patients were assessed. Among the 119 cases in which there was a preliminary diagnosis by a non-sarcoma pathologist, there were 37 modifications in diagnosis (31.1%) by the sarcoma pathologist, resulting in 17 (14.2%) modifications in management. Among the 110 cases in which molecular tests were performed, there were 29 modifications in diagnosis (26.4%) through the genomic results, resulting in 12 (10.9%) modifications in management. Our study confirms that expert pathology review is of utmost importance for optimal sarcoma diagnosis and management and should be assisted by molecular methods in selected cases.

## 1. Introduction

Sarcomas represent a group of rare tumors of soft tissues and bones, characterized by remarkable heterogeneity in their histopathology, molecular profile, and clinical behavior. They constitute about 15% of cancers in children and less than 1% of cancers in adults. More than 120 subtypes were included in the 2013 WHO classification of tumors of soft tissue and bone [1,2], whereas further refinement mainly based on molecular characteristics resulted in approximately 170 different entities, both malignant and of intermediate malignancy [3,4]. Most of these subtypes have distinct histological, molecular, and clinical features. Towards this direction, the 2020 WHO classification of tumors of soft tissue and bone included a few new entities that can be diagnosed exclusively using molecular methods, such as sarcomas with neurotrophic tyrosine receptor kinase (*NTRK*) fusions [5] and BCL6 corepressor (*BCOR*) alterations [6].

Sarcomas exhibit a wide range of genetic alterations, such as chromosomal translocations involving transcription factors, complex chromosomal aberrations, overexpression of receptor kinase ligands, inactivation of regulatory proteins, gene mutations, and gene amplifications. In total, approximately 30–40% of sarcomas are characterized by a well-defined recurrent genetic alteration that contributes to their pathogenesis [7,8,9,10]. These genetic alterations were initially detected using traditional molecular methods, such as polymerase chain reaction (PCR) and fluorescence in situ hybridization (FISH). In addition to these techniques, genome sequencing is increasingly used today for the confirmation of the diagnosis of a specific fraction of sarcomas.

Given their rarity and heterogeneity, there is a certain degree of complexity within sarcomas’ classification. Accurate diagnosis is crucial for patients’ management to enable a histology-tailored approach, as recommended both in surgery [11] and systemic therapy for sarcomas [12], requiring, thus, an expert pathologist. There is a number of studies that have demonstrated a rate of discrepancy between initial and expert diagnosis in sarcoma up to 50% [13,14,15,16,17,18,19,20,21], whereas the effect of pathology review on management has scarcely been investigated [19]. These studies took place mainly in USA and Europe, with some recent reports also from India [15], Australia [19] and Canada [20]. However, their retrospective design (except for a French study [22]) results to selection bias, as only difficult cases were included. 

Furthermore, molecular biology is a valuable tool to refine the diagnosis of subtypes known to harbor a genetic aberration [23,24,25]. Several retrospective series of selected cases [26,27,28], as well as a limited number of large retrospective studies including different sarcoma histotypes [29,30] and a population-based study in 3 European region [24], have demonstrated the significance of molecular methods in sarcoma diagnosis. There is only one prospective study investigating the impact of molecular methods in six sarcoma histotypes diagnosis and management [23].

The current prospectively designed study focuses on the value of expert pathology reviews and the application of molecular analysis to new sarcoma cases in Greece over a 2.5-year period. To this end, an in-house next-generation sequencing (NGS) test specific for sarcoma diagnosis was used, coupled with traditional molecular methods upon indication. We assessed the effect of expert pathology review, combined with molecular diagnostics, on the management of sarcoma patients.

## 2. Materials and Methods

### 2.1. Study Design and Patients

This is a prospective, multicentre, observational study including 13 centers of the Hellenic Group of Sarcoma and Rare Cancers (HGSRC). The aim of the study is to assess the medical impact of histopathological classification by two sarcoma pathologists, followed by genetic aberration detection, on the diagnosis of soft tissue and bone sarcomas. Study inclusion criteria were: initial biopsy or surgical specimen (before any preoperative therapy) with histological diagnosis of soft tissue or bone sarcoma of any stage; available tissue (paraffin block) for molecular studies; age ≥18 years; and access to patient’s clinical record. The initial goal was to include 100 patients.

Demographic and clinical data were collected, including patient’s sex and age at diagnosis, subtype of sarcoma, tumor localization (upper extremities, lower extremities, retroperitoneum, abdomen/pelvis, head and neck, trunk wall, gynecological, viscera, other), depth of the tumor (for extremities sarcoma, superficial or deep), and stage (local, locally advanced, metastatic, or recurrent).

### 2.2. Procedures

The work flow of this study is shown in Figure 1. The physicians of the participating centers (medical oncologist, surgical oncologist, or radiation oncologist) recommended the inclusion of the patients (Figure 1) based on the initial histological diagnosis (probable or certain sarcoma diagnosis). In all cases, diagnosis was made through a preoperative biopsy, surgical resection of the tumor, or both. In some patients, an initial pathological examination was performed by an external pathologist not specializing in sarcoma (preliminary diagnosis). In these cases, tissue (paraffin block +/− slides) was transferred to one of the two sarcoma pathologists (G.A., D.P.) of the HGSRC for revision. Clinical information as well as imaging data were also provided to the pathologist. Histological diagnosis after expert review, prior to molecular testing, was recorded (expert diagnosis). The diagnosis was based on histopathological features and immunohistochemistry (IHC). At this time, the referring physician was informed of the report of the expert pathologist and was asked to complete the first part of the Case Report Form (CRF-A), including information about the therapeutic strategy (proposed management in terms of primary surgery, (neo)-adjuvant chemotherapy, and/or radiotherapy) based on the expert diagnosis. Histological reports and CRF-A were sent to the study coordinator (S.K.). Finally, the specimen was sent to the Laboratory of Genetics of the Saint-Savvas Cancer Hospital in Athens (E.D., L.M.) for molecular testing. The results of the molecular analysis were sent to the study coordinator as well as to the corresponding expert pathologist, who placed the final diagnosis. The referring physician was informed of the final diagnosis after the molecular analysis and then asked to complete the second part of the CRF (CRF-B), informing them whether the therapeutic strategy was modified (primary surgery, (neo)-adjuvant chemotherapy, or/and radiotherapy). Histological reports and CRF-B were sent to the study coordinator, who analyzed the data.

### 2.3. Molecular Test

Molecular analysis was performed in the Laboratory of Genetics of the Saint-Savvas Cancer Hospital in Athens. One paraffin block from each specimen was selected, and approximately 50 mg of tumor tissue (2–3 sections of 5 μm from the surgical specimen or 4 sections of 10 μm from the biopsies) was removed. Total RNA extraction was performed using NucleoSpin total RNA FFPE XS (Macherey Nagel, Duren, Germany). Approximately 100–200ng of this material was used in reverse transcription reactions following specific Libraries synthesis with the Ion AmpliSeq Library Kit 2.0 (Thermo Fisher Scientific, Waltham, MA, USA) and a specific NGS panel designed in-house for sarcoma diagnosis. The NGS test was performed on an Ion GeneStudio S5 System on samples from all patients included in the study. This sarcoma fusion panel comprises 86 different chimeric transcripts, representing the majority of genetic aberrations in soft tissue and bone tumors (Table 1).

Apart from the NGS (sarcoma fusion panel), when the sarcoma pathologist suspected a well-differentiated liposarcoma (WDLPS) or dedifferentiated liposarcoma (DDLPS), FISH was also performed for the detection of *MDM2* and *CDK4* amplification. Furthermore, DNA NGS for *KIT*/*PDGRα* point mutations was performed in cases of suspected diagnosis of gastrointestinal stromal tumor (GIST) and real-time PCR (RT-PCR) to detect β-catenin mutations in case of desmoid tumor. Finally, in case of suspicion of undifferentiated small round cell sarcoma, negative for Ewing sarcoma family of tumor translocations, *BCOR* alterations were also tested using RT-PCR.

### 2.4. Statistics

This is a proof-of-concept study conducted to investigate the added value of systematic molecular testing upon sarcoma diagnosis in Greece. Descriptive statistics were used to calculate the frequency of genetic aberrations in the study population. The percentage of cases with diagnosis modification and the percentage of cases with therapeutic strategy modification following molecular analysis were also calculated. In addition, we evaluated the percentage of cases with diagnosis modification and the percentage of cases with therapeutic strategy modification following an expert pathology review.

## 3. Results

### 3.1. Patient Characteristics

Between 1 January 2020 and 31 May 2023, 128 specimens from 122 patients were assessed in this study, including 75 males (61.5%) and 47 females, who were enrolled by nine public and four private hospital centers/clinics and comprised the full analysis set (Appendix A). From six patients, both initial biopsy and surgical specimens were included in the study, leading to 128 specimens. Eight additional specimens from seven patients were deemed ineligible because the paraffin block never arrived or was not reviewed by the expert pathologists, and a molecular test was not performed. Thus, these specimens were not included in the analysis. At enrolment, the patients’ median age was 54 years (18–83). Table 2 shows the distribution of the different histotypes based on the initial histological diagnosis. In 119 cases (93%), the initial diagnosis was made by a non-sarcoma pathologist (preliminary diagnosis). When two or more diagnoses were considered by the initial pathologist, only the most probable one is indicated in Table 2. Adipocytic tumors (*n* = 24) were the most prevalent diagnostic group, followed by tumors of uncertain differentiation (*n* = 22) and fibroblastic/myofibroblastic tumors (*n* = 20). Among the enrolled patients, myxoid liposarcoma and undifferentiated pleomorphic sarcoma (UPS) were the most prevalent histotypes, represented by 8 cases each. Soft tissue sarcoma comprised the vast majority of the study population, with only 6 (4.7%) chondro-osseous tumors.

Patients’ samples were obtained by biopsy in 48 cases (37.5%) or surgical resection of the primary tumor in 65 cases (46.1%), whereas in the remaining 15 cases, the information was not available. If surgical resection of the primary tumor was already performed, eventual modifications in patient management were assessed in terms of adjuvant therapies. The stage of the disease at study entry was classified as local in 65 cases (50.8%), locally advanced in 18 cases (14.1%), and metastatic in 34 cases (26.6%). Nine additional tumors (7.0%) were considered local recurrences, while one patient presented with multiple benign lesions. The stage was not available in the remaining one case (0.8%). The primary tumor was located in the lower extremities (*n* = 46, 35.9%), the trunk wall (*n* = 23, 18%), the viscera (*n* = 11, 8.6%), the upper extremities (*n* = 12, 9.4%), the retroperitoneum (*n* = 10, 7.8%), the head and neck (*n* = 4, 3.1%), the uterus (*n* = 2, 1.6%), or other rare sites (*n* = 5, 3.9%), such as the testicle. Ten (7.8%) sarcomas are located in the bones. In the remaining three patients (2.3%), the primary location was not available.

### 3.2. Effect of Expert Pathology Review

Among the 119 cases in which there was a preliminary diagnosis by a non-sarcoma pathologist, 38 had minor discordances (identical diagnosis of connective tissue tumor but different grade or subtype), 18 had major discordances (misdiagnosis of sarcoma, final diagnosis of a benign lesion, or very significant impact on the management), whereas the remaining 63 had full concordance (Table 3). In particular, there were 37 modifications in diagnosis (31.1%) by the sarcoma pathologist (Table 4), resulting in 17 (14.2%) modifications in management. DDLPS was the most common subtype that was misdiagnosed by the initial pathologist. Clinical implications of diagnostic modification included a different chemotherapy regimen for advanced disease (*n* = 9), avoidance of adjuvant treatment (*n* = 4), wider surgical excision (*n* = 2), adjuvant radiation therapy (*n* = 1), and a completely different management for thyroid carcinoma metastasis instead of primary sarcoma (*n* = 1).

In addition, the diagnoses of 19 cases (9.2%) were refined without modification in management. For instance, three tumors, initially described as liposarcoma, were finally classified as DDLPS (*n* = 2) and myxoid LPS (*n* = 1). Another case of osteosarcoma was subclassified as extraskeletal chondroblastic osteosarcoma. Three more cases diagnosed as “sarcoma” and one as “low grade tumor” were subclassified as myxofibrosarcoma (MFS) (*n* = 3) and malignant peripheral nerve sheath tumor (MPNST) (*n* = 1). In the remaining cases, additional features were revealed by the sarcoma pathologist, leading to diagnostic refinement, including tumor grade and morphology of the sarcoma cells.

### 3.3. Effect of Molecular Test

Of the 128 samples, 110 were sent to the Laboratory of Genetics for molecular testing, whereas the remaining 18 samples were never sent due to logistic issues. Of these, 51 samples were obtained by surgical excision, 35 by biopsy, whereas in the remaining 24 cases, the information was not available. The specific sarcoma NGS test (RNA sequencing) was performed in the majority of the specimens (*n* = 84), generating 1 non-evaluable, 14 positive, and 69 negative results. In 17 additional cases, the RNA sample volume was insufficient for NGS analysis. Lastly, RNA NGS was not performed in 9 cases, in which the pathologist opted for another molecular test (FISH for *MDM2* amplification: *n* = 7, RT-PCR for *CTTNB1* mutation: *n* = 1, DNA NGS for *GNAS* point mutation: *n* = 1). In 11 samples, in addition to NGS, a second molecular method was performed (FISH for *MDM2* amplification: *n* = 4, FISH for *FUS*-*DDIT3* fusion: *n* = 1, RT-PCR for *BCOR* rearrangement: *n* = 4, DNA NGS panel for *KIT*/*PDGFRa* point mutations: *n* = 2). Positive NGS results involved most frequently the *EWSR1* gene in Ewing sarcoma/PNET (*n* = 5), the *FUS* gene in myxoid LPS (*n* = 4), followed by *COL1A1* (dermatofibrosarcoma protuberans), *STAT6* (solitary fibrous tumor), *NTRK3* (NTRK-rearranged sarcoma), *NR4A3* (extraskeletal myxoid chondrosarcoma), and the *USP6* gene (nodular fasciitis), with one case each of these genes. FISH was performed in total in 14 cases (13 for *MDM2* amplification and 1 for *FUS*-*DDIT3* fusion), generating 1 non-evaluable, 9 positive, and 4 negative results. Other molecular tests (RT-PCR for *BCOR* rearrangement, *CTTNB1* mutation, *KIT*/*PDGFRa* mutation, and DNA NGS for *GNAS* mutation) generated 2 positive and 6 negative results.

Overall, we detected the presence of a genetic abnormality in 25 of the 110 tested specimens (22.7%), whereas 17 specimens were deemed unsuitable for testing, and the results were considered non-evaluable in another 2 cases. In 29 patients (26.4%), the genomic findings (positive or negative result) were found to be diagnosis-changing (Table 5), resulting in 12 modifications in the patient’s management (10.9%). A molecular test was performed on a bioptic specimen in 10 cases and on a surgical specimen in 19 cases. Patients with an initial differential diagnosis including Ewing’s sarcoma family of tumors, derived the most benefit from molecular methods (*n* = 8), followed by patients initially diagnosed with adipocytic tumors (*n* = 6). The clinical effects of molecular genetic testing involved different regimens for advanced disease (*n* = 9), different follow-ups (*n* = 2), and different decisions for adjuvant radiation therapy (*n* = 1).

## 4. Discussion

This study evaluated the clinical and therapeutic impact of the expert pathology review and the additional molecular diagnostics in an unselected series of new soft tissue and bone sarcoma cases in Greece. The purpose of our study was to prospectively assess the impact of these two interventions on both sarcoma diagnosis and management, while previous studies were mainly retrospective and focused only on diagnosis. We found that expert assessment led to diagnostic modification in 30.3% of the cases, with an effect on treatment strategy in 13.4%. This is the first attempt to systematically review all new sarcoma cases in Greece, following the founding of the HGSRC in 2018, a national network of sarcoma experts. Thenceforth, efforts are made to manage sarcoma patients within the HGSRC.

Our findings are in accordance with previous reports. The importance of second opinion in sarcoma diagnostics has been recognized since more than 30 years ago [13,14,15,22,31]. A population-based study within 3 European regions, which analyzed retrospectively 1463 sarcoma patients, revealed a modification in diagnosis in more than 40% of cases as a result of a centralized expert pathology review [16]. A discordance rate of 34% was reported by an older study of 216 STS and bone sarcomas in the USA [17]. A similar rate of 37% was reported by another study in the USA [18]. These studies assessed the effect on diagnostic accuracy and reported discordances involving mainly histological grade and subtype. This was also observed in our study, with several correct initial diagnoses missing histological grade. Histological grading by sarcoma pathologists is highly valid and reproducible, as shown by a Japanese study of adult STS cases [32]. A recent retrospective Australian study reported a change in diagnosis in 21.3% of sarcoma cases that benefited from an external review, with an effect on treatment strategy in 6.6% [19]. A Canadian study of STS initially diagnosed in a general anatomical pathology service and reviewed in North America over a 10-year period revealed partial agreement in 42.5% and zero agreement in 14.4% of the cases [20]. A higher concordance of 77.6% is reported by the Swiss Sarcoma Network [21].

The molecular diagnosis contradicted the pathological diagnosis in 26.4% of our unselected sarcoma patients. Several retrospective studies have evaluated the contribution of molecular investigations in sarcoma diagnosis in the past and refer to ancillary molecular tests (FISH, RT-PCR, etc.), with varying effects on diagnostic classification and clinical management [26]. Auxiliary molecular examinations consisting of FISH (for most of the known sarcoma fusions) associated or not with targeted RNA sequencing, led to diagnostic refinement or revision in 32% of a selected series of 84 uncommon, unclassifiable mesenchymal tumors in a study led by international expert sarcoma pathologists [27]. Importantly, in this study, 17% of the tumors harbored a recently described genetic aberration, highlighting the evolving landscape of sarcoma molecular pathology. Clinical implications were noted in only 6% of the cases. The only prospective data on the clinical effect of molecular methods in sarcoma diagnosis and management are reported by the French Sarcoma Group [23]. Indeed, in this study, molecular analysis using FISH, comparative genomic hybridization, quantitative PCR, or RT-PCR resulted in diagnosis modification in 13.8% of the case mix, which included six sarcoma types with known genetic aberrations. A meta-analysis of 70 studies evaluating molecular analysis in the diagnosis and prediction of the prognosis of STS demonstrated that FISH for *MDM2* amplification, RT-PCR for *SYT*-*SSX* fusion, and *CTNNB1* mutation were useful for WDLPS/DDLPS, synovial sarcoma, and desmoid tumors, respectively [33].

Traditional molecular methods, such as FISH and PCR, certainly play an important role in diagnosis refinement, as illustrated by the example of *MDM2* amplification in WDLPS and DDLPS. NGS-based techniques have emerged over the last decade as robust diagnostic tools for the detection of pathognomonic sarcoma fusion transcripts [34,35]. Anchored multiplex PCR, for instance, has been proven practical for routine diagnosis [36]. Several retrospective studies on the implementation of comprehensive genomic profiling through NGS in the management of sarcoma patients have been published recently, mainly from Comprehensive Cancer Centers in the USA. Gounder et al. reported a refinement or reassignment of 10.5% of sarcoma diagnoses through NGS [29]. The application of NGS technology improved sarcoma diagnosis, enabling the classification of 15/64 unclassified samples from Chinese STS patients [28]. We recently showed that the use of NGS led to diagnosis modification in 9.1% of sarcoma patients enrolled in a large retrospective study across multiple European institutions [30]. The therapeutic relevance of diagnostic modification based on molecular methods is poorly reported in the literature and warrants further evaluation in prospective studies. Importantly, both the pathologist and the clinician should be aware that not all sarcoma cases are suitable for NGS/fusion analysis. The sarcoma histomorphology should be taken into serious consideration for patient selection.

In our study, the proportion of patients harboring a genomic alteration was relatively small (22.7%), whereas a higher proportion of approximately 40% is described for STS. This is probably explained by the relatively small sample size and the histotypes that were included. The diagnostic value of molecular methods has been demonstrated for certain histotypes, including synovial sarcoma [37], rhabdomyosarcoma [10,38], clear cell sarcoma [39], and desmoid tumors [40]. The numbers of these histotypes, as well as GIST (*n* = 2), are underrepresented in our cohort. In addition, although the in-house diagnostic NGS was performed on the vast majority of the samples, the suitable molecular method for each sample was not always employed. Importantly, in 7/11 cases of expert diagnosis of DDLPS, no FISH was performed (the sample was insufficient in one case and for organizational reasons in the remaining 6 cases). Similarly, RT-PCR for the *CTNNB1* mutation was used in one out of two desmoid tumor samples.

The most common diagnostic modification referred to was DDLPS, which was misdiagnosed by the initial pathologist. FISH for *MDM2* amplification detection is an established tool for the classification of adipose tissue tumors [41]. This analysis is highly recommended in retroperitoneal or other tumors that could correspond to a LPS based on imaging and histological criteria. A subgroup of retroperitoneal high-grade UPS lacking histologic evidence of lipoblasts turns out to be DDLPS by molecular analysis [42]. Furthermore, molecular biological analysis of 331 LPS from the Netherlands led to reclassification in approximately 25% of cases, enabling us to better distinguish WDLPS with or without DDLPS from myxoid LPS and pleomorphic LPS [43]. This can lead to a different management approach involving the use of radiation therapy or drugs. Accordingly, a negative *MDM2* result can distinguish a lipoma from a WDLPS, preventing an unnecessary wider surgical excision in some cases, as was the case in one of our patients.

Specific translocations’ detection for Ewing’s sarcoma family of tumors is also a valuable diagnostic tool [7,44]. Once the diagnosis of Ewing sarcoma is made, a special intensive chemotherapy protocol is used, given the aggressiveness of these tumors and their sensitivity to chemotherapy. In our study, the use of NGS led to the confirmation of Ewing sarcoma for patients who were treated with the appropriate chemotherapy regimen. On the contrary, NGS enabled the exclusion of this diagnosis in three patients, who were ultimately diagnosed with another sarcoma subtype and treated accordingly, avoiding this toxic chemotherapy. Furthermore, three patients were finally diagnosed with “undifferentiated small round cell sarcoma” (USRC), a new entity whose optimal treatment strategy is yet to be defined [45]. It should be noted that during the last few years, novel translocations have been identified that are specific for this family of tumors, such as *CIC*-*DUX4* [46] and *BCOR*-rearrangement [6].

Molecular methods are extremely important for patients with GIST. Mutations in *KIT* or *PDGFRα* occur in almost 85% of GIST, serving as a diagnostic tool. However, these mutations play a more important role for the clinician, as they have a high predictive value of sensitivity to treatment; mutations in exon 11 of *KIT* predict a higher sensitivity to imatinib than mutations in exon 9 of *KIT* [47]. Moreover, these mutations also have a prognostic value, with mutations in exon 18 of the *PDGFRα* gene indicating a better prognosis than mutations in exon 11 of *KIT*, which have a better prognosis than mutations in exon 9 of *KIT*. These elements have been incorporated in guidelines’ algorithms of GIST management after surgery, regarding the indication or not of adjuvant treatment by imatinib [48,49,50].

Genomic findings can also inform the prognosis of different STS histotypes. For instance, synovial sarcoma with *SYT*-*SSX1* translocation has been associated with a worse prognosis compared to patients with *SYT*-*SSX2* one in various studies [9,51,52,53,54], although it has not been confirmed in some others [55,56]. Clinical trials evaluating more intensive treatment protocols in these patients are warranted. It should be noted that a 67-gene expression signature was established, which predicts the clinical outcome of localized sarcomas, in order to inform clinical decisions on adjuvant treatment [57].

The current study was not designed to explore the occurrence of clinically actionable targets in sarcoma patients, given the NGS platform employing RNA sequencing for the detection of the most common sarcoma fusions. Nevertheless, the detection of *ETV6*-*NTRK3* gene fusion had a significant therapeutic implication, as we treated this patient with the NTRK inhibitor larotrectinib instead of cytotoxic chemotherapy. NGS has also been used over the last few years to detect pharmacologically tractable alterations in sarcoma patients [29,58,59]. Apart from GIST, only a minority of sarcomas harbor a driver targetable genetic aberration, including *ALK* fusion in IMT targeted with crizotinib and other TKIs [60], *PDGFb* fusion in dermatofibrosarcoma protuberans (DFSPs) targeted with imatinib [61], and *CDK4* amplification in WDLPS and DDLPS targeted with CDK4/6 inhibitors [62]. NTRK inhibitors are new potent tissue-agnostic targeted drugs that were recently evaluated in basket trials with tumors harboring *NTRK* rearrangements [63,64]. However, the frequency of *NTRK* fusions is extremely low in adult STS, and specific histological criteria have been proposed for testing [65].

To summarize, NGS (or other molecular testing) is recommended mainly in sarcoma diagnostics when the specific pathological diagnosis is not certain or the clinical presentation is uncommon. In addition, in cases of suspicion of a histotype known to harbor a recurrent genetic aberration (such as WDLPS/DDLPS, the Ewing sarcoma family of tumors, or DFSP), NGS can be used to detect this highly specific diagnostic biomarker and lead to diagnostic refinement. Finally, molecular tests are indicated if the result might have prognostic or predictive implications. This is the case for subtypes with available targeted therapies, such as NTRK-rearranged sarcomas, IMT with ALK fusion, and GIST.

There were some challenges while conducting this trial. We noted some important delays in obtaining the final pathology report, incorporating the molecular results. Therefore, in some cases, the clinician decides the treatment plan before the results of the molecular methods. The sample from 17 patients, which originated mainly from biopsies, was not sufficient to perform molecular tests. Inadequate biopsies were another challenge observed by the pathologists, preventing exhaustive IHC testing. Finally, a false positive result of the detection of *EWSR1*-*FLI1* misled the clinicians and pathologists until the repeat of NGS, since the molecular findings were not in line with the histomorphology.

## 5. Conclusions

The precise diagnosis of sarcomas is the cornerstone for planning the treatment strategy for these patients and requires expert pathological assessment, given the rarity and heterogeneity of these tumors. The molecular pathogenesis of many sarcoma histotypes has been rigorously characterized. There is an increasing body of retrospective evidence on the clinical relevance of genetic aberrations in sarcomas in terms of diagnostic accuracy, with potential therapeutic implications. Prospective data on the added value of the implementation of molecular methods in sarcoma management were reported from the French Sarcoma Group. In this study, we prospectively investigated the contribution of both expert pathology review and molecular biology in sarcoma diagnosis and management. We noted a discordance rate of 31.1% between the initial and expert diagnosis, with an effect on clinical management in 14.2% of cases. The results of the molecular methods were indispensable for the final diagnosis in 26.4% of the samples. We conclude that expert pathology assessment is the mainstay of optimal sarcoma diagnosis and management, whereas molecular testing requested by the expert pathologist and the multi-disciplinary board can be very helpful in certain cases.

## Figures and Tables

**Figure 1 cancers-16-02314-f001:**
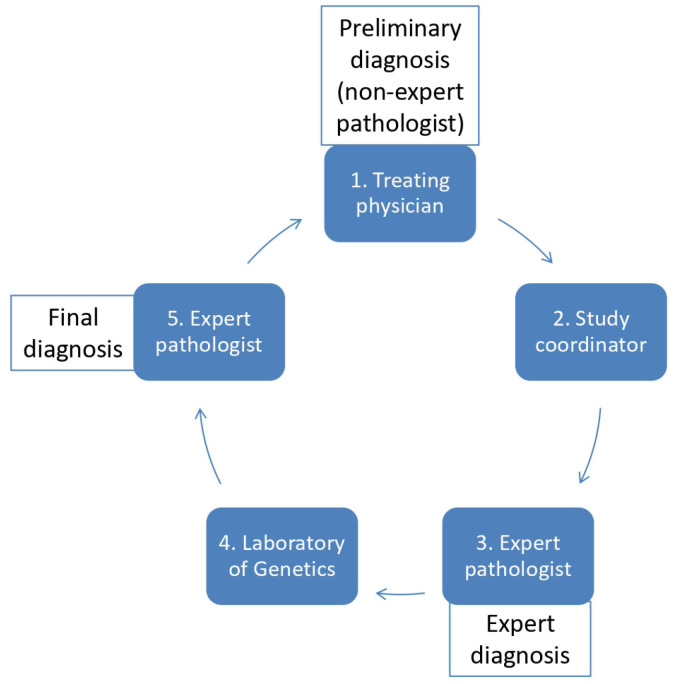
Study design.

**Table 1 cancers-16-02314-t001:** Next-generation sequencing fusion panel used for sarcoma diagnosis.

Tumor	Fusion Gene
EWING SARCOMA	EWSR1-FLI1
	EWSR1-ERG
	EWSR1-FEV
	EWSR1-ETV1
	EWSR1-ETV4
	FUS-FEV
	FUS-ERG
	EWSR1-SMARCA5
EWING-LIKE SARCOMA	EWSR1-NAFTC2
	NAFTC2-EWSR1
	EWSR1-SP3
	CIC-DUX4l1
	CIC-FOXO4
SYNOVIAL SARCOMA	SS18-SSX1
	SS18-SSX4
	SS18L1-SSX1
ALVEOLAR RHABDOMYOSARCOMA	PAX3-FOXO1
	FOXO1-PAX3
	PAX7-FOXO1
	PAX3-NCOA2
	PAX3-NCOA1
ALVEOLAR SOFT PART SARCOMA	TFE3-ASPCR1
	ASPCR1-TFE3
INFINTILE FIBROSARCOMA	NTRK3-ETV6
	ETV6-NTRK3
LIPOMA	LPP-HMGA2
	HMGA2-LPP
LIPOBLASTOMA	COL1A2-PLAG1
	COL3A1-PLAG1
ENDOMETRIAL STROMAL SARCOMA	JAZF1-SUZ12
	MEAF6-PHF1
SOFT TISSUE MYOEPITHELIAL TUMOR/CARCINOMA	EWSR1-ZNF444
MESENCHYMAL CHONDROSARCOMA	HEY1-NCOA2
DESMOPLASTIC SMALL ROUND CELL TUMOR	EWSR1-WT1
PERICYTOMA	ACTB-GLI1
	GLI1-ACTB
DERMATOFIBROSARCOMA PROTUBERANS	COL1A1-PDGFB
EXTRASKELETAL MYXOID CHODROSARCOMA	EWSR1-NR4A3
	TFG-NR4A3
INFLAMATORY MYOFIBROBLASTIC TUMOR	CLTC-ALK
	ATIC-ALK
	TPM3-ALK
MYOEPITHELIOMA	EWSR1-PBX1
SPINDLE AND ROUND CELL SARCOMA	EWSR1-PATZ1
ANGIOMATOID FIBROUS HISTIOCYTOMA	EWSR1-ATF1
	ATF1-EWSR1
	EWSR1-CREB1
	FUS-ATF1
MYXOID LIPOSARCOMA	FUS-DDIT3
	DDIT3-FUS
	EWSR1-DDIT3
LOW GRADE FIBROMYXOID SARCOMA	FUS-CREB3L1
	FUS-CREB3L2
	CREB3L2-FUS
	EWSR1-CREB3L1
	EWSR1-CREB3L2
ANEURYSMAL BONE CYST	COL1A1-USP6
	CDH11-USP6
	OMD-USP6
	THRAP3-USP6
	CNBP-USP6

**Table 2 cancers-16-02314-t002:** Distribution of the initial histological diagnosis.

Diagnostic Category	Initial Diagnosis	N
ADIPOCYTIC TUMOURS	Lipoma	2
	ALT	1
	WDLPS	6
	LPS	3
	DDLPS	4
	myxLPS	8
FIBROBLASTIC/MYOFIBROBLASTIC TUMOURS	cellular angiofibroma	1
	desmoid tumor	5
	DFSP	2
	myxoinflammatory fibroblastic sarcoma	1
	Low-grade FMS	3
	MFS	3
	SFT	5
VASCULAR TUMOURS	EHE	4
	AS	4
SMOOTH MUSCLE TUMOURS	Leiomyoma	2
	uLMS	1
	LMS	5
SKELETAL MUSCLE TUMOURS	RMS	4
GIST	GIST	2
CHONDRO-OSSEOUS TUMOURS	Chordoma	1
	CS	2
	OS	3
PERIPHERAL NERVE SHEATH TUMOURS	MPNST	8
TUMOURS OF UNCERTAIN DIFFERENTIATION	atypical ossifying fibromyxoid tumor	1
	atypical fibroxanthoma	1
	CCS	3
	ES	3
	SS	3
	EMC	1
	ASPS	1
	UPS	8
	undifferentiated sarcoma	1
USRC SARCOMAS OF BONE AND SOFT TISSUE	Ewing-like	4
OTHER SARCOMA	sinonasal biphenotypic sarcoma	1
	dendritic cell sarcoma	1
	myxoid sarcoma	1
	sarcoma	5
OTHER LESIONS	low-grade tumor	1
	myxoid spindle cell lesion	1
	sex cord tumor (v. sclerosing SS)	1
	benign lesion	3
	NA	8

ALT: atypical lipomatous tumor, AS: angiosarcoma, ASPS: alveolar soft part sarcoma, CCS: clear cell sarcoma, CS: chondrosarcoma, DDLPS: dedifferentiated liposarcoma, DFSP: dermatofibrosarcoma protuberans, EHE: epithelioid hemangioendothelioma, EMC: extraskeletal myxoid chondrosarcoma, ES: epithelioid sarcoma, FMS: fibromyxoid sarcoma, GIST: gastrointestinal stromal tumor, (u)LMS: (uterine) leiomyosarcoma, LPS: liposarcoma, MFS: myxofibrosarcoma, MPNST: malignant peripheral nerve sheath tumor, myxLPS: myxoid liposarcoma, OS: osteosarcoma, RMS: rhabdomyosarcoma, SFT: solitary fibrous tumor, SS: synovial sarcoma, UPS: undifferentiated pleomorphic sarcoma, WDLPS: well-differentiated liposarcoma.

**Table 3 cancers-16-02314-t003:** Discordance in diagnosis between initial and expert pathologists.

	No Discordance	Minor Discordance	Major Discordance
Number	63	38	18
%	52.9	31.9	15.1

**Table 4 cancers-16-02314-t004:** Cases with diagnosis modification by expert pathology review.

Initial Diagnosis	Expert Diagnosis	Modification in Management
AS	undifferentiated spindle-cell Sa G3	different CT for metastatic disease
atypical fibroxanthoma	pleomorphic dermal sarcoma	wider excision
benign	FMS G1	wider excision
DDLPS	UPS G2	no (surgery + adjuvant RT)
desmoid tumor	myofibroblastic tumor	no (surgery already performed)
desmoid tumor	fibrous dysplasia	no (active surveillance decided)
Ewing-like	ASPS	avoidance of adjuvant CT
Ewing-like	dedif. CS, mesench.CS, or small cell OS	no (CT for metastatic disease)
fibromatosis	UPS or DDLPS	yes (CT for inoperable disease)
interdigitating dendritic cell Sa	MPNST G2	avoidance of adjuvant CT
lipoma	ES	tazemetostat instead of CT for metastatic disease
LMS	DDLPS with MFS G3 differentiation	no (surgery only)
MPNST	DDLPS	no (surgery + adjuvant RT)
MPNST	dermatic melanocytoma	no
MPNST, RMS, or Ewing Sa	Ewing Sa	Ewing-type CT for metastatic disease
myx spindle cell lesion	FMS G1 or MFS G1	no (adjuvant RT)
myxLPS	DDLPS	no
myxoinflammatory fibroblastic Sa	MFS G2	no
myxSa	DDLPS	no
OS	sarcomatoid mesothelioma	different CT for advanced disease
OS	CS G2	avoidance of adjuvant CT
RMS	mesenchymal neoplasm of myoblastic differentiation	no (surgery + adjuvant CT, RT)
RMS	MPNST G2	different CT for metastatic disease
sarcoma or spindle cell Ca	intimal Sa	no
sex cord tumor (Sertoli) or sclerosing SS	sclerosing epithelioid FS	different CT for advanced disease
SFT	MPNST G2	no (surgery + adjuvant RT)
SFT	epithelioid AS	different CT for inoperable disease
SS	meta from thyroid Ca	completely different management
SS	URCS (SS or Ewing)	no
undifferentiated Sa	pleomorphic lipoma	avoidance of adjuvant treatment or wider excision
UPS	LMS or DDLPS	no (CT for metastatic disease)
UPS	epithelioid AS	different CT for metastatic disease
UPS	MFS G3	no
UPS	LMS G3	no (surgery + adjuvant RT)
WDLPS	DDLPS	no (CT for metastatic disease)
WDLPS	probaly myxLPS of breast	no

AS: angiosarcoma, ASPS: alveolar soft part sarcoma, Ca: carcinoma, CT: chemotherapy, CS: chondrosarcoma, DDLPS: dedifferentiated liposarcoma, ES: epithelioid sarcoma, FMS: fibromyxoid sarcoma, FS: fibrosarcoma, LMS: leiomyosarcoma, MFS: myxofibrosarcoma, MPNST: malignant peripheral nerve sheath tumor, myxLPS: myxoid liposarcoma, OS: osteosarcoma, RMS: rhabdomyosarcoma, RT: radiation therapy, Sa: sarcoma, SFT: solitary fibrous tumor, SS: synovial sarcoma, UPS: undifferentiated pleomorphic sarcoma, URCS: undifferentiated round-cell sarcoma, WDLPS: well-differentiated liposarcoma.

**Table 5 cancers-16-02314-t005:** Cases with diagnosis modification by the molecular results.

Expert Diagnosis	Molecular Result	Molecular Diagnosis
ALT	FISH *MDM2* neg	lipoma
atypical ossifying fibromyxoid tumor	*ESR1*-*FLI1* (first time), NGS neg (repeat)	no modification (false positive)
CCS	*ESR1*-*ATF1*	angiomatoid fibrous histiocytoma
DDLPS or myxLPS	FISH *MDM2* pos	DDLPS with myxoid component
dedif. CS, or mesench.CS, or small cell OS	NGS neg	probably small cell OS
ES	*BCOR*-*CCNB3*	no modification
Ewing-like	NGS neg	URCS
Ewing-like	NGS neg, *BCOR* neg	URCS
Ewing-like	*ESR1*-*FLI1*	Ewing Sa
Ewing-like or ASPS	NGS neg	ASPS
Ewing-like or MPNST	*EWSR1*-*FLI1*	Ewing Sa
Ewing-like, MPNST, or SS	NGS neg	MPNST
Ewing-like or myoepithelial Ca	NGS neg, *BCOR* neg	URCS
Ewing-like or spindle-cell RMS	NGS neg, *BCOR* neg	RMS
high-grade Sa	FISH *MDM2* pos	DDLPS
lipoma or ALT	NGS neg, FISH *MDM2* pos	ALT
LMS	*ETV6*-*NTRK3*	*NTRK*-rearranged LMS
LMS or DDLPS	NGS neg	LMS
LMS or GIST	NGS neg, *KIT*/*PDGFRa*/*BRAF*/*RAS* neg	LMS
LMS or GIST	*MET*exon14 mut, *KIT*exon10 mut, *TP53*exon5 mut, *PDGFRa*/*BRAF*/*RAS* neg	LMS
MFS G1 or intramuscular myxoma	DNA seq *GNAS* neg	MFS G1
MPNST or SS	NGS neg	MPNST
myofibroblastic tumor	*MYH9*-*USP6*	nodular fasciitis
myxLPS or MFS	NGS neg, FISH *FUS*-*DDIT3* neg	MFS
myxLPS or MFS G1	NGS neg	MFS G1
probaly myxLPS	NGS neg, FISH *MDM2* pos	WDLPS
SS	*TAF15*-*NR4A3*	EMC
UPS or DDLPS	NGS neg, FISH *MDM2* neg	UPS

ALT: atypical lipomatous tumor, ASPS: alveolar soft part sarcoma, Ca: carcinoma, CCS: clear cell sarcoma, CS: chondrosarcoma, DDLPS: dedifferentiated liposarcoma, EMC: extraskeletal myxoid chondrosarcoma, ES: epithelioid sarcoma, FISH: fluorescence in situ hybridization, GIST: gastrointestinal stromal tumor, LMS: leiomyosarcoma, MFS: myxofibrosarcoma, MPNST: malignant peripheral nerve sheath tumor, myxLPS: myxoid liposarcoma, NGS: next-generation sequencing, OS: osteosarcoma, RMS: rhabdomyosarcoma, Sa: sarcoma, SS: synovial sarcoma, UPS: undifferentiated pleomorphic sarcoma, URCS: undifferentiated round-cell sarcoma, WDLPS: well-differentiated liposarcoma.

## Data Availability

The original data are available on request from the authors.

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
