# Peer review of "The Impact of Expert Pathology Review and Molecular Diagnostics on the Management of Sarcoma Patients: A Prospective Study of the Hellenic Group of Sarcomas and Rare Cancers"

_cancers, 2024, doi:10.3390/cancers16132314_

Round 1

Reviewer 1 Report

Comments and Suggestions for Authors

I have read this study with great interest, it underlines the necessity of reference pathologic review of all soft tissue sarcomas. However there are some minor remarks which should be modified before possible publication:

1.    Authors should present in the table the results of the concordance of the primary diagnosis

-          I suggest the following division: - complete discordance (misdiagnosis of soft tissue sarcoma, final diagnosis of a benign lesion, or very significant impact on the choice of treatment), - change of diagnosis within STS subtypes (without significant impact on final treatment), - full agreement in diagnosis

2.  I suggest to better describe the indications for screening NGS test.

3. Moreover, I doubt if GIST should be included in this analysis as usually it is diagnosed without any problems with IHC.

4. Gene names should be written consistently in Italics.

Author Response

Reviewer 1

I have read this study with great interest, it underlines the necessity of reference pathologic review of all soft tissue sarcomas. However there are some minor remarks which should be modified before possible publication:

  1. Authors should present in the table the results of the concordance of the primary diagnosis

      I suggest the following division: - complete discordance (misdiagnosis of soft tissue sarcoma, final diagnosis of a benign lesion, or very significant impact on the choice of treatment), - change of diagnosis within STS subtypes (without significant impact on final treatment), - full agreement in diagnosis

Thank you very much for your comment. We added a new table 3, dividing all cases that were initially seen by a non expert pathologist into these 3 groups (major discordance, minor discordance, no discordance).

We also added the following phrase in the text (results section, 3.2 effect of expert pathology review):

Among the 119 cases in which there was a preliminary diagnosis by a non-sarcoma pathologist, “38 had minor discordances (identical diagnosis of connective tissue tumor but different grade or subtype), 18 had major discordances (misdiagnosis of sarcoma, final diagnosis of a benign lesion, or very significant impact on the management), whereas the remaining 63 had full concordance (Table 3).”

  1. I suggest to better describe the indications for screening NGS test.

We added the following paragraph in the discussion section, to summarize the indication for NGS/molecular tests in sarcoma patients:

To summarize, NGS (or other moleculr test) is recommended mainly in sarcomas diagnostics, when the specific pathological diagnosis is not certain or the clinical pres-entation is uncommon. In addition, in case of suspicion of a histotype known to harbor a recurrent genetic aberration (such as WDLPS/DDLPS, Ewing sarcoma family of tumors, and DFSP), NGS can be used to detect this highly specific diagnostic biomarker and lead to diagnostic refinement. Finally, molecular tests are indicated if the result might have prognostic or predictive implication. This is the case for subtypes with available tar-geted therapies, such as NTRK-rearranged sarcomas, IMT with ALK fusion and GIST.”

  1. Moreover, I doubt if GIST should be included in this analysis as usually it is diagnosed without any problems with IHC.

We included these 2 cases of GIST, as we included consecutive sarcoma patients. We had decided to include GIST in this trial, as in some cases the diagnosis can be challenging (differential diagnosis with leiomyosarcoma); we have some examples in our registry.

  1. Gene names should be written consistently in Italics.

Thank you for your remark, this is done.

Reviewer 2 Report

Comments and Suggestions for Authors

the article is original and well written and adds information to the literature on a difficult, broad and ever-changing topic. well played

Comments on the Quality of English Language

excellent grammar, just some corrections in punctuation

Author Response

Reviewer 2

the article is original and well written and adds information to the literature on a difficult, broad and ever-changing topic. well played

Comments on the Quality of English Language

excellent grammar, just some corrections in punctuation

Thank you very much for your positive comments.

Reviewer 3 Report

Comments and Suggestions for Authors

Dear authors,

I am sorry, but I did not find the purpose of this study. It seems that you have merely explained a procedure that is already well-established. There appears to be a lack of novelty in this research.

Author Response

Reviewer 3

Dear authors,

I am sorry, but I did not find the purpose of this study. It seems that you have merely explained a procedure that is already well-established. There appears to be a lack of novelty in this research.

Thank you for your comment.

We designed this prospective study, which we believe is important, for the following reasons:

-Although the effect of expert pathology review in sarcomas has been well reported by different studies (In European countries, USA, Japan), leading to the recommendation that new sarcoma cases should be reviewed by sarcoma pathologists, a large proportion of these studies are retrospective (selective only difficult cases of second opinion). In addition, data is completely lacking from countries like Greece, where there are not recognized sarcoma units. This is the first attempt to systematically review all new sarcoma cases in our country by 2 sarcoma pathologist and we think that it is important to publish the results.

  -In addition, we prospectively evaluated the effect of molecular tests in sarcoma diagnosis and management. The recommendations are not very clear, with some countries performing molecular test in every patient with a diagnosis of a sarcoma histotype known to harbor a genetic aberration (even if histological diagnosis is certain) and others performing it upon request of the pathologist. The evidence of the effect of molecular methods is based almost exclusively on retrospective studies, with the exception of the French Gensarc study. Most studies evaluated the effect of molecular methods on sarcoma diagnosis only and not on management. Our study is prospective evaluating both the effect of expert pathology review and our in-house sarcoma NGS panel.

Round 2

Reviewer 3 Report

Comments and Suggestions for Authors

Dear authors,

1. Kindly add a section of Related work where you can present some of the previous studies by other researchers for Sarcoma all over the world.

2. It would be great to add a subsection of Contribution/Purpose of the study so that the readers easily understand. 
